Captivity causes taxonomic and functional convergence of gut microbial communities in bats

Xiao Yanhong xiaoyh767@nenu.edu.cn 1
Xiao Guohong 1
Liu Heng 1
Zhao Xin 1
Sun Congnan 1
Tan Xiao 1
Sun Keping 1
Liu Sen 2
Feng Jiang fengj@nenu.edu.cn 1 3
1 Jilin Provincial Key Laboratory of Animal Resource Conservation and Utilization, Northeast Normal University , Changchun , Jilin , China
2 Institute of Resources & Environment, Henan Polytechnic University , Jiaozuo , Henan , China
3 College of Life Science, Jilin Agricultural University , Changchun , Jilin , China
Harrison Xavier
Electronic publication date: 2019 Apr 30
Publication date: 2019
Volume: 7
Electronic Location ID: e6844
Received 2018 Dec 12; Accepted 2019 Mar 18
Copyright: ©2019 Xiao et al.
Copyright year: 2019
Copyright holder: Xiao et al.
License: This is an open access article distributed under the terms of the Creative Commons Attribution License, which permits unrestricted use, distribution, reproduction and adaptation in any medium and for any purpose provided that it is properly attributed. For attribution, the original author(s), title, publication source (PeerJ) and either DOI or URL of the article must be cited.
License URL: https://creativecommons.org/licenses/by/4.0/

Keywords: Diet, Microbiome, Convergence, Bat

Funding: National Natural Science Foundation of China 31700319 31870354 Fundamental Research Funds for the Central Universities 2412017QD026 China Postdoctoral Science Foundation 2018M631852 This work was supported by the National Natural Science Foundation of China (Grant No. 31700319, 31870354), Fundamental Research Funds for the Central Universities (Grant No. 2412017QD026) and Project funded by China Postdoctoral Science Foundation (Grant No. 2018M631852). The funders had no role in study design, data collection and analysis, decision to publish, or preparation of the manuscript.

==============================
Background

Diet plays a crucial role in sculpting microbial communities. Similar diets appear to drive convergence of gut microbial communities between host species. Captivity usually provides an identical diet and environment to different animal species that normally have similar diets. Whether different species’ microbial gut communities can be homogenized by a uniform diet in captivity remains unclear.

Methods

In this study, we compared gut microbial communities of three insectivorous bat species (Rhinolophus ferrumequinum, Vespertilio sinensis, and Hipposideros armiger) in captivity and in the wild using 16S rDNA sequencing. In captivity, R. ferrumequinum and V. sinensis were fed yellow mealworms, while H. armiger was fed giant mealworms to rule out the impact of an identical environment on the species’ gut microbial communities.

Results

We found that the microbial communities of the bat species we studied clustered by species in the wild, while the microbial communities of R. ferrumequinum and V. sinensis in captivity clustered together. All microbial functions found in captive V. sinensis were shared by R. ferrumequinum. Moreover, the relative abundances of all metabolism related KEGG pathways did not significantly differ between captive R. ferrumequinum and V. sinensis; however, the relative abundance of “Glycan Biosynthesis and Metabolism” differed significantly between wild R. ferrumequinum and V. sinensis.

Conclusion

Our results suggest that consuming identical diets while in captivity tends to homogenize the gut microbial communities among bat species. This study further highlights the importance of diet in shaping animal gut microbiotas.

Introduction

Trillions of microorganisms reside in animal guts, and these microorganisms constitute the animal’s gut microbiota, which is important for animal health (Flint et al., 2012). Ley et al. (2008a) found that animals who were closely related taxonomically had more similar gut microbial compositions. Phylogenetic congruence of microflora communities and their hosts was also observed among bat families (Ingala et al., 2018; Phillips et al., 2012). These studies indicated that host evolutionary history strongly impacts gut microbiome compositions. Although gut microbial communities are host-specific, they can be influenced by the host’s diet, developing immune system, chemical exposures and initial colonizers (Donaldson, Lee & Mazmanian, 2015). Diet has been suggested to have the greatest impact on microbiota assembly (Donaldson, Lee & Mazmanian, 2015). Diet shapes the gut microbial community by providing substrates that differentially support or enhance the growth of specific microbes (De Filippo et al., 2010; Scott et al., 2013; Wang et al., 2013). Taxonomic compositions of the gut microbial communities of different host species with similar diets appeared to converge in some studies (Carrillo-Araujo et al., 2015; Delsuc et al., 2014; Muegge et al., 2011). Ley et al. (2008a) also found that animals with similar diets (i.e., herbivores, carnivores, omnivores) had more similar gut microbiome compositions.

Wild animals in captivity are usually housed under uniform conditions that include identical diets and environments (Hale et al., 2018). This represents a rapid and dramatic dietary and environmental change to the animals. The gut microbiome has been reported to rapidly respond to an altered diet (David et al., 2013). However, whether different species’ gut microbiomes will respond similarly to the uniform conditions of captivity remains uncertain. A study comparing the gut microbial diversity in two woodrat species in the wild and in captivity found that the microbial communities in these species did not converge (Kohl, Skopec & Dearing, 2014). Principal coordinate analysis results showed that the microbial signatures of the captive woodrats still clustered by species (Kohl, Skopec & Dearing, 2014). Woodrats are herbivores, which represents only one mammalian dietary type. Carnivores represent another important dietary type, of which, insectivores are thought to represent the ancestral condition for placental mammals (O’Leary et al., 2013). However, whether the taxonomic compositions of different insectivorous species’ microbial gut communities tend to converge under identical dietary and environmental conditions remains unclear. In addition, the two wood rat species that Kohl, Skopec & Dearing (2014) studied were closely related. Given the phylogenetic distance among the hosts in the present study, it is unclear whether a homogenous diet/environment or the host’s evolutionary history more strongly impacts the microbiome community composition.

Bats (order Chiroptera) are the second largest mammalian group (Wilson & Reeder, 2005). Most bats are insectivores, which is also thought to be the ancestral condition for bats (Dawson & Krishtalka, 1984). To determine whether diet/environment or evolutionary more strongly impacts the microbiome, we sampled feces (guano) from three bat species from three families (Rhinolophidae, Vespertilionidae and Hipposideridae): the greater horseshoe bat (Rhinolophus ferrumequinum), the Asian parti-colored bat (Vespertilio sinensis) and the great Himalayan leaf-nosed bat (Hipposideros armiger) in the wild and in captivity. We then compared the bacterial communities in both the wild and captive samples between these three species. We captured bats in the wild, brought them back to the laboratory and housed them in identical environments but provided different food. Rhinolophus ferrumequinum and V. sinensis were fed the same food (yellow mealworms), while H. armiger were provided giant mealworms, thus forming a comparison to eliminate the impact of environment on the gut microbiome. Given that diet strongly influences microbiome composition and similar diets appear to drive convergence of gut microbial communities between host species (Delsuc et al., 2014; Muegge et al., 2011), we predicted that the gut microbiome compositions of captive R. ferrumequinum and V. sinensis under identical environmental and dietary conditions would converge with each other but would differ from captive H. armiger. In addition, taxonomy and function are decoupled in microbial ecosystems (Graham et al., 2016; Inkpen et al., 2017; Louca, Parfrey & Doebeli, 2016). Microbial functions may converge despite the microbial community’s taxonomic compositions varying among host species (Phillips et al., 2017). Thus, microbial functions were also predicted and compared among different bats species both in the wild and in captivity to investigate whether the gut microbiome function converges in the captive bats.

Material and Methods

Field sampling of bats

All three bat species are insectivores. Rhinolophus ferrumequinum feeds preferentially on lepidopterans, particularly the noctuid species, which constitute approximately 41% of the bat’s diet (Jones, 1990). The bats also eat coleopterans, which constitute approximately 33% of their diet, of which, dung beetles and cockchafers are often consumed (Jones, 1990). The dietary composition of V. sinensis mainly comprises Lepidoptera (mean relative percentage: 32.8%), Diptera (27.5%) and Coleoptera (22.6%), but the proportion of each order varies seasonally (Fukui & Agetsuma, 2010). H. armiger’s diet mainly comprises 31.59–37.21% Coleoptera and 15.38–22.87% Lepidoptera (Han & He, 2012).

Eight greater horseshoe bats, seven Asian parti-colored bats and eight great leaf- nosed bats were collected from Jilin, Heilongjiang and Guizhou, China, respectively, during the summer of 2018. Bats were collected from one group of each species. Fecal samples were collected from these bats in the field. During the summer of 2017, we collected 10 greater horseshoe bats from three groups, of which, three bats were from Jilin, one from Liaoning, and six from Shannxi, China. Ten Asian parti-colored bats in one group and 10 great leaf- nosed bats in one group were collected from Heilongjiang and Shannxi, China, respectively, during the summer of 2017. These bats were returned to the laboratory, and different bat species were housed in separate cages for 4–6 months before collecting their fecal samples. Details on the bats collected are shown in Table 1.

Table 1 Summary of samples included in this study.

Sample type	Species	Number	Sex	Age	Weight(g; mean ± SD)	Forearm length(mm; mean ± SD)	Site	
Wild	R. ferrumequinum	8	3M +5F	Adults	18.44 ± 1.41	60.55 ± 0.92	Jilin	
	V. sinensis	7	F	Adults	21.89 ± 3.60	49.44 ± 2.12	Heilongjiang	
	H. armiger	8	M	1 Juvenile + 7 Adults	67.03 ± 9.01	95.82 ± 2.58	Guizhou	
Captive	R. ferrumequinum	10	1M +9F	6 Juveniles + 4 Adults	25.84 ± 5.39	60.83 ± 1.35	Jilin/Liaoning/ Shannxi	
	V. sinensis	10	F	Adults	21.33 ± 3.83	50.87 ± 1.21	Heilongjiang	
	H. armiger	10	8M + 2F	Adults	69.12 ± 8.08	95.64 ± 3.38	Shannxi	

Collection of fecal samples

Fecal samples were used because dietary signals in the microbiome are more easily detected in fecal samples than in intestinal samples (Ingala et al., 2018). Bats were captured in the field using mist nets placed at cave entrances, immediately recovered from the nets, and placed in separate clean holding bags to await processing. We recorded each bat’s sex, age, weight, forearm length, and reproductive condition (Table 1 and Data S1). Feces were collected directly from the bottom of the holding bags and placed in sterile tubes using sterile forceps, then stored in dry ice before transport to the laboratory. The bags were checked frequently to ensure the samples’freshness. In the laboratory, the greater horseshoe bats and the Asian parti-colored bats were fed yellow mealworms (Tenebrio molitor), while the great leaf-nosed bats were fed giant mealworms (Zophobas morio) for comparison. We kept the bats for 4–6 months, and collected their fecal pellets less than 15 min after defecationin the laboratory. Each bat’s sex, age, weight, forearm length, and reproductive condition was recorded (Table 1 and Data S1), then the bats were placed in separate clean cages, which were placed on sterile brown paper. Feces were collected from the brown paper and placed in sterile tubes, then temporarily stored in liquid nitrogen. The brown paper was checked frequently to ensure the feces’ freshness. All samples were stored in −80 °C until DNA extraction.

Sampling was conducted with permission from the local forestry department. The National Animal Research Authority of Northeast Normal University, China (approval number: NENU-20080416) and the Forestry Bureau of Jilin Province, China (approval number: [2006]178) approved all study protocols.

DNA extraction

Fifty-three fecal samples were used, including 23 from the wild bats and 30 from the captive bats. DNA was extracted from all fecal samples using the E.Z.N.A.®Stool DNA Kit (Omega Bio-Tek, Inc., Norcross, GA, USA) per the manufacturer’s instructions and stored at −20 °C for further analysis. Extracted DNA was measured using a NanoDrop NC2000 spectrophotometer (Thermo Fisher Scientific, Waltham, MA, USA) and agarose gel electrophoresis to estimate DNA quantity and quality, respectively.

16S rDNA amplicon pyrosequencing

The V3-V4 region of the bacterial 16S rRNA genes were amplified via PCR using the forward primer, 338F (5′-ACTCCTACGGGAGGCAGCA-3′), and the reverse primer, 806R (5′-GGACTACHVGGGTWTCTAAT-3′) (Dennis et al., 2013). Sample-specific 7-bp barcodes were incorporated into the primers for multiplex sequencing. The PCR components contained 5  µl of Q5 reaction buffer (5 ×), 5 µl of Q5 High-Fidelity GC buffer (5 ×), 2 µl of dNTPs (2.5 mM), 1  µl of each forward and reverse primer (10 µM), 0.25  µl of Q5 High-Fidelity DNA polymerase (5 U/µl), 2  µl of DNA template, and 8.75  µl of ddH2O. The PCR conditions consisted of initial denaturation at 98 °C for 2 min, followed by 25 denaturation cycles at 98 °C for 15 s, annealing at 55  °C for 30 s, extension at 72 °C for 30 s, and a final extension at 72 °C for 5 min. PCR products were purified with Agencourt AMPure Beads (Beckman Coulter, Indianapolis, IN, USA) and quantified using the PicoGreen dsDNA Assay Kit (Invitrogen, Carlsbad, CA, USA). The individual PCR products were then pooled in equal amount, and sequenced using the paired-end 2 ×300 bp method on the Illumina MiSeq platform with MiSeq Reagent Kit v3 at Shanghai Personal Biotechnology Co., Ltd. (Shanghai, China). All raw sequences were deposited into the NCBI Sequence Read Archive under accession numbers SRR8238420–SRR8238472.

Sequence analysis

Sequencing data were processed using the Quantitative Insights Into Microbial Ecology (QIIME, v1.8.0) (Caporaso et al., 2010). Briefly, raw sequences with unique barcodes were assigned to respective samples. Sequences shorter than 150 bp, having average Phred scores of <20, containing ambiguous bases, or sequences containing more than 8-bp mononucleotide repeats were regarded as low-quality sequences and removed (Chen & Jiang, 2014; Gill et al., 2006). Paired-end reads were assembled using FLASH (Magoč & Salzberg, 2011). Assembled sequences were trimmed of barcodes and sequencing primers. After chimera detection, the remaining trimmed and assembled sequences were clustered into operational taxonomic units (OTUs) at 97% sequence identity using UCLUST (Edgar, 2010). A representative sequence was selected from each OTU using default parameters. Representative sequences were aligned to the Greengenes Database (DeSantis et al., 2006) using the best hit (Altschul et al., 1997) to classify the taxonomy, which was conducted using BLAST. An OTU table was then generated to record each OTU’s abundance per sample and the OTU’s taxonomy. OTUs containing less than 0.001% of the total sequences across all samples were discarded. To minimize the differences in sequencing depth across samples, an averaged, rounded, rarefied OTU table was generated by averaging 100 evenly resampled OTU subsets under 90% of the minimum sequencing depth for further analysis.

Bioinformatics and statistical analysis

Sequence data were mainly analyzed using QIIME v1.8.0 and R v3.2.0. Beta diversity was analyzed to investigate the microbial communities’ structural variation across samples using UniFrac distance metrics (Lozupone & Knight, 2005; Lozupone et al., 2007) and visualized via principal coordinate analysis (PCoA) and nonmetric multidimensional scaling (NMDS) (Ramette, 2007). UniFrac is the only distance metric that considers the phylogenetic relationships between microorganisms, and UniFrac-based beta diversity has become a standard analytic method in microbiome studies. Therefore, we also chose the UniFrac distance to characterize the community structure in our study. Differences in the UniFrac distances for pairwise comparisons among groups were determined using Student’s t-test and the Monte Carlo permutation test with 1,000 permutations, then visualized using box-and-whiskers plots. For UniFrac distance-based pairwise comparisons among groups, we used a very conservative Bonferroni post-hoc correction method to perform the multiple corrections and evaluate the significance of the comparison. Permutational multivariate analysis of variance (PERMANOVA) (McArdle & Anderson, 2001) and analysis of similarities (ANOSIM) (Clarke, 1993; Warton, Wright & Wang, 2012) were conducted using the R package “vegan” (v1.6-9) (Oksanen et al., 2005) to assess the significance of the differentiation of the microbiota structures among groups. A Venn diagram was generated to visualize the shared and unique OTUs among groups using the R package “VennDiagram” (v2.4-6) (Chen & Boutros, 2011) based on the occurrence of OTUs across groups regardless of their relative abundance (Zaura et al., 2009). Microbial functions were predicted using Phylogenetic Investigation of communities by Reconstruction of Unobserved States (PICRUSt, v1.0.0) (Langille et al., 2013) in the Kyoto Encyclopedia of Genes and Genomes (KEGG) database (Kanehisa et al., 2004) based on high-quality sequences. The relative abundances of predicted functions in each sample were calculated based on the abundance matrix obtained via PICRUSt, and significant differences in each function’s relative abundances among different species were tested using analysis of variance (ANOVA) or the Kruskal–Wallis test (Wallis, 1952). Results were considered significant at p < 0.05.

Results

Sequencing results

A total of 768,990 and 1,466,150 16S rDNA sequences were obtained from the microbiomes of the 23 wild and 30 captive bats, respectively, and the average sequence numbers per sample were 33,434 and 48,872, respectively. Rarefaction analysis demonstrated that the sequencing depth was sufficient for each sample (Fig. S1). A total of 3,504 and 7,057 OTUs were recovered at the similarity clustering threshold of 97%.

Shared microbial species were increased in captive bats

Venn diagrams were plotted to visualize the shared and unique OTUs (roughly equivalent to bacterial species) among three species of wild and captive bats. The captive bats we sampled shared more OTUs than did the wild bats (Fig. 1). A total of 2,022 OTUs (approximately 29% of the total OTUs) were shared by the three species in captivity, but only 228 OTUs (approximately 7% of the total OTUs) were shared by the wild bats. Approximately 71% of the OTUs from captive V. sinensis and R. ferrumequinum were shared, but only 18% were shared by these two species in the wild. The proportions of OTUs shared by V. sinensis and H. armiger were approximately 39% and 12% in captivity and the wild, respectively. Minimal difference was noted between the proportions of shared OTUs in the captive and wild R. ferrumequinum and H. armiger, of which, the proportions were nearly 36% and 29%, respectively.

Figure 1 Venn diagram of shared and unique OTUs in the fecal bacterial communities of three bat species.

(A) Wild bats’ fecal samples. (B) Captive bats’ fecal samples. WFVs, WFRf and WFHa represent fecal samples from V. sinensis, R. ferrumequinum and H. armiger collected from the wild respectively. FVs, FRf and FHa represent fecal samples from captive V. sinensis, R. ferrumequinum and H. armiger respectively.

Microbial compositions converged in captive bats fed the same food

A NMDS based on unweighted beta diversity values indicated that the gut microbial communities in the wild bat were clustered strongly by bat species (Fig. 2A). However, the gut microbial community clustering was altered in the captive bats (Fig. 2B). In the captive bats, the gut microbial communities of the bats fed the same food (i.e., V. sinensis and R. ferrumequinum fed yellow mealworms) clustered together, while the gut microbial communities of H. armiger, fed giant mealworms, clustered alone. A PCoA based on unweighted UniFrac distances also demonstrated similar clustering results using NMDS based on unweighted UniFrac distances. In the wild bats, PC1, PC2 and PC3 accounted for nearly 42% of the variation, and samples were separated roughly by bat species (Fig. S2A). In the captive bats, PC1, PC2 and PC3 accounted for 48% of the variation in microbial composition (Fig. S2B).

Figure 2 Wild and captive bats’ fecal bacterial communities clustered using nonmetric multidimensional scaling analysis of the unweighted UniFrac distance matrix.

Wild (A) and captive (B) bats’ fecal bacterial communities clustered using nonmetric multidimensional scaling analysis. Each point corresponds to a fecal sample colored according to bat species with different symbols corresponding to host family (red circle, Hipposideridae, green square, Vespertilionidae, blue triangle, Rhinolophidae).

Analyzing the differences in the UniFrac distances for pairwise comparisons among groups revealed that the differences between each pair group were significant in the wild samples (Fig. S3A, Table 2), while the differences between V. sinensis and R. ferrumequinum were not significant in the captive samples (p = 0.381 and 0.085) (Fig. S3B, Table 2). However, the differences between H. armiger and the other two species were all significant in the captive samples (Fig. S3B, Table 2). Statistical analyses of the significance of the differentiation in the microbiota structure among the groups also yielded similar results. The differences among groups were significant both for the wild and the captive bats (all p ≤ 0.001, Table 3). However, PERMANOVA and ANOSIM analyses cannot assess the significance of the differentiation between pairwise groups when more than two groups are analyzed. Thus, we did not find that the differences between V. sinensis and R. ferrumequinum were not significant in the captive samples based on the PERMANOVA and ANOSIM analysis results.

Table 2 Results of Student’s t-test and the Monte Carlo permutation test of differences in the UniFrac distances for pairwise comparisons among groups.

	Group 1	Group 2	t statistic	p-valuea	
Wild	All within Group	All between Group	−17.881	0.000***	
	WFVs vs. WFVs	WFVs vs. WFRf	−16.090	0.000***	
	WFVs vs. WFVs	WFVs vs. WFHa	−22.230	0.000***	
	WFRf vs. WFRf	WFVs vs. WFRf	−8.363	0.000***	
	WFRf vs. WFRf	WFRf vs. WFHa	−7.793	0.000***	
	WFHa vs. WFHa	WFVs vs. WFHa	−12.663	0.000***	
	WFHa vs. WFHa	WFRf vs. WFHa	−8.920	0.000***	
Captive	All within Group	All between Group	−19.425	0.000***	
	FVs vs. FVs	FVs vs. FRf	−2.499	0.381	
	FVs vs. FVs	FVs vs. FHa	−15.337	0.000***	
	FRf vs. FRf	FVs vs. FRf	−3.014	0.085	
	FRf vs. FRf	FHa vs. FRf	−15.644	0.000***	
	FHa vs. FHa	FVs vs. FHa	−48.674	0.000***	
	FHa vs. FHa	FHa vs. FRf	−58.733	0.000***	
Notes.

a p-value was corrected by Bonferroni method.

*** p-value ≤ 0.001.

Table 3 Statistical analyses accessing significance of differentiation of microbiota structure among groups.

	Results of PERMANOVA analysis	
	Df	Sums of Sqs	F. Model	r2	p-value	
Wild	2	2.139	4.565	0.313	0.001***	
Captive	2	2.656	8.320	0.381	0.001***	
	Results of ANOSIM analysis	
	R statistic	p-value	Number of permutationos	
Wild	0.873	0.001***	999	
Captive	0.803	0.001***	999	
Notes.

*** p-value ≤ 0.001.

Convergence of microbial function in captive bats fed the same food

Finally, we predicted the microbial functions of wild and captive bats using PICRUSt, which yielded 5,971 and 4,771 KEGG pathways, respectively. Venn diagrams showed that in total 5,495 KEGG pathways were shared among the wild bat samples, and 3,964 were shared among the captive bat samples (Fig. 3). Unlike the wild bats, one hundred percent of the microbial functions in captive H. armiger were shared by the other two species, and all microbial functions in captive V. sinensis were shared by R. ferrumequinum. Thus, in terms of presence/absence, the microbial functions appeared converged in the captive bats.

Figure 3 Venn diagram of shared and unique microbial functions in three bats species.

(A) Wild bats. (B) Captive bats. WFVs, WFRf, WFHa, FVs, FRf and FHa are defined in the legend of Fig. 1.

Moreover, in terms of the relative abundance of functions, we found that the relative abundances of all metabolism-related KEGG pathways did not significantly differ between captive R. ferrumequinum and V. sinensis, while the relative abundance of “Glycan Biosynthesis and Metabolism” differed significantly between the wild R. ferrumequinum and V. sinensis (Fig. 4). In addition, except “Glycan Biosynthesis and Metabolism”, “Enzyme Families” and “Biosynthesis of Other Secondary Metabolites”, no significant differences were found in the relative abundances of any other metabolic pathways among the three bat species in the wild (Fig. 4A), while the relative abundances of all metabolism-related KEGG pathways except “Metabolism of Cofactors and Vitamins”, “Lipid Metabolism” and “Carbohydrate Metabolism”, differed significantly between captive H. armiger and the other two bat species (Fig. 4B). This result indicated that the microbial functions converge in the captive bats fed the same food in terms of the relative abundance of functions.

Figure 4 The relative abundance of microbial functions related to metabolism predicted by PICRUSt.

(A)Wild bats. (B) Captive bats. WFVs, WFRf, WFHa, FVs, FRf and FHa are defined in the legend of Fig. 1. Different letters in each KEGG pathway indicate significant differences (p < 0.05) in the relative abundances of this function in different bats species.

Discussion

In this study we investigated the influence of identical diets under laboratory conditions on the gut microbial communities of three insectivorous bat species. Feces are sampled as a proxy for the gut microbiome in many studies of wild mammal microbiomes (Hale et al., 2018; Kohl, Skopec & Dearing, 2014; Phillips et al., 2017; Sommer et al., 2016). Moreover, more signals from the host’s diet are retained in fecal samples than in intestinal samples (Ingala et al., 2018). Thus, in our study we compared the microbial communities from fecal samples of three captive bat species, as well as the fecal microbial communities of their conspecific bats in the wild. The microbiome compositions of the bats in our study were mainly composed of Proteobacteria and Firmicutes, which occupied more than 80% of the microbiome (Figs. S4 and S5). This was consistent with previous work on bat microbiomes (Carrillo-Araujo et al., 2015; Ingala et al., 2018; Phillips et al., 2017; Phillips et al., 2012).

Comparing the microbial communities of fecal samples from three bat species in the wild revealed that the microbial signatures of R. ferrumequinum, V. sinensis and H. armiger in the wild cluster by species when measured by principal coordinate analysis. Microflora communities of wildlife species are shaped by complex processes, including host phylogeny, dietary strategy and reproductive conditions (Phillips et al., 2012). Though R. ferrumequinum, V. sinensis and H. armiger are all insectivores, wild bat diets are varied, species-specific and belong to different bat families. Thus, taxonomic compositions of gut microbial communities differ among bat species in the wild. Our result was consistent with the study of Phillips et al. (2017).

Comparing the microbial communities in fecal samples from three captive bats species showed that the microbial compositions of two bat species (R. ferrumequinum and V. sinensis) fed the same food converged markedly, while they differed from those of H. armiger fed different food. This result highlighted the importance of diet on gut microbial communities. Diet shapes the gut microbiota by providing substrates that differentially support or enhance specific microbial growth (De Filippo et al., 2010; Scott et al., 2013; Wang et al., 2013). The gut microbiota can in turn enable their host to adapt to new dietary niches (Ley et al., 2008a). In this study, captive bats were fed mealworms, which is a novel and high-quality diet for the previously wild bats. Long-term dietary intake influences the structure and activity of gut microbiota (Duncan et al., 2007; Ley et al., 2006; Muegge et al., 2011; Walker et al., 2010; Wu et al., 2011). After 4–6 months’ feeding in the laboratory, the bats’ gut microbiotas should have adapted to the new diet. Feeding the bats the same food means that same substrates are provided to the gut microbiota; thus, the gut microbial signatures of captive R. ferrumequinum and V. sinensis should cluster together. Similar results were obtained in a study comparing the gut microbiotas of captive colobine monkeys. This study found that the gut microbial communities were more similar in the colobine species who consumed the same diet (Hale et al., 2018). In contrast, captive H. armiger were fed a different diet (giant mealworms) than were R. ferrumequinum and V. sinensis, and the gut microbial communities of captive H. armiger did not converge with the other two bat species. This result eliminated the impact of environment on the gut microbial communities because the three bat species were housed in identical environments in the laboratory. Further, this suggested that identical diets contribute to microbial community convergence in various bat species. However, fecal samples usually include bacteria that are ingested with the food (e.g., the commensal bacteria in the mealworms), and distinguishing these bacteria from the host-derived bacteria in the fecal microbiome is difficult. Thus, the gut microbiota in this study did not specifically refer to the host-derived bacteria. The microbial community compositions between species fed uniform diets may have converged due to changes in the compositions of the host-associated gut microbiome or a much larger shared component of the fecal microbiome based on a completely shared diet comprising mealworms and their commensal bacteria or both. Our results highlight the need for future studies to address this issue, for example, incorporating dietary classifications via metabarcoding and classifying the microbiomes of the invertebrate prey. In addition, our results differed from those obtained by Kohl, Skopec & Dearing (2014), whose results suggested that microbial communities of various woodrat species clustered by species rather than converged together after being exposed to similar diets. The bat species in our study were insectivores, while the woodrat species in the work of Kohl, Skopec & Dearing (2014) were herbivores. Herbivorous digestive systems contain multiple enzymes originating from different microbial species needed to process (hemi)celluloses, lignin-derivatives and insoluble starches, thus supporting a highly diverse ecosystem (Karasov, Martínezdel Rio & Caviedes-Vidal, 2011). Moreover, bacterial diversity increases as the host diet diverges from carnivorous to omnivorous to herbivorous in mammalian guts (Ley et al., 2008b). Thus, we hypothesize that the higher bacterial diversity in herbivorous mammals allows them to retain more species-specific microbial communities in captivity than do mammals that eat animal-based diets. In addition, more unique bacteria make it difficult for microbial communities of different herbivorous species to cluster together although they have similar diets. Kohl et al.’s (2014) indicating that 64% and 51% of OTUs were retained in the two captive woodrat species studied may support our hypothesis. Another possible explanation for the differences between the bats in this study and the woodrats in Kohl et al.’s study (2014) may be that the difference between the diets of captive bats and those in nature is larger than that of the woodrats. In xenarthrans (anteaters, sloths, and armadillos), especially myrmecophagous mammals (i.e., mammals that eat termites and ants), the effect of captivity on their gut microbiomes is especially noticeable in animals whose diets differ markedly in captivity and in nature (Superina, 2011).

The divergence or convergence of microbial community compositions differs from the divergence or convergence of their microbial functions. Different combinations of microbial lineages may achieve comparable community functions, meaning that microbial communities may differ in taxonomic composition but be similar in function (Phillips et al., 2017). In terms of their presence/absence, the unique microbial functions of the bats we studied were lost when the bats were taken from the wild into captivity, especially in H. armiger and V. sinensis. This may be due to the lack of a different environment or similar nutrient compositions in the mealworm larvae (Rumpold & Schlüter, 2013) provided as food in the laboratory. However, in terms of the function frequency, we found that the relative abundances of most metabolic functions were similar, although the microbial community compositions differed among the three bat species in the wild. This finding was similar to that of Phillips et al. (2017) and supports the hypothesis of functional redundancy in the gut ecosystem, which is defined as functions conferred by multiple bacteria that can be shared across both related and unrelated bacterial species (Moya & Ferrer, 2016). That is, although the microbial composition varies, different microbiotas may perform similar functions (Pérez-Cobas et al., 2013). Comparing the relative abundances of metabolic functions among the three bat species in captivity, the relative abundances of most metabolic functions were similar between R. ferrumequinum and V. sinensis but differed in H. armiger. Combining the result that the microbial compositions of R. ferrumequinum and V. sinensis converged together but diverged from H. armiger, we inferred that the different foods led to metabolic tuning of microbial functions and the identical diets which in captivity led to the convergence of both microbial compositions and microbial functions.

Unexpectedly, the number of OTUs found in the bats’ fecal samples was increased in the captive bats compared with those in the wild. In other words, the gut microbiota was more diverse in the captive bats than in the wild bats. This was surprising because the number of OTUs in the gut microbiome was expected to have been greatly decreased due to the single-food-source diet and captive conditions. This observation also contradicted the findings of previous studies on the influence of captivity on the animal microbial communities, which found that bringing animals into captivity resulted in a loss of microbial diversity (Redford et al., 2012; Kohl, Skopec & Dearing, 2014; Kueneman et al., 2016). Several points may explain this. First, the bat species may have been exposed to each other’s microbes due to their being in captivity in the same laboratory; thus, the overall community became more diverse because microbes were shared among species. Second, the bacteria ingested with the mealworms may have increased the diversity of the captive bats’ gut microbial communities. Although the microbial diversity was increased in the captive bats, the microbial function types were decreased in our study, indicating that selection by host diet primarily acts on metagenomic functions. Further research is required to investigate the possible reasons for this.

Conclusions

Comparing the results from PCoA and NMDS analyses between wild and captive bats suggests that the identical diets that were provided in captivity contributed to the taxonomy convergence of the gut microbial communities of R. ferrumequinum and V.  sinensis. In addition, in terms of functional level, the identical diets while in captivity yielded more similar relative abundances of metabolic functions in the gut microbiomes of captive R. ferrumequinum and V. sinensis than in the wild bats, indicating that the identical diet while in captivity contributed to the convergence of the gut microbial community functions. Finally, the gut microbial diversity was surprisingly higher in the captive bats than in the wild bats. However, understanding why this phenomenon occurred requires further study. This study highlights the diet’s crucial role in shaping captive bat gut microbiotas.

Supplemental Information

Figure S1 Rarefaction curves for OTUs defined at 97% similarity per bat species

(A) Wild bats (B) Captive bats. Points are means ± SE, with the numbers of bats per group shown in Table 1. The meanings of WFVs, WFRf, WFHa, FVs, FRf and FHa are same as in Fig. 1, (see Fig. 1 legend).

Click here for additional data file.

Figure S2 Wild and captive bats’ fecal bacterial communities clustered using principal coordinates analysis of the unweighted UniFrac distance matrix

Wild (A) and captive (B) bats’ fecal bacterial communities clustered using principal coordinates analysis. Each point corresponds to a fecal sample colored according to bat species with different symbols corresponding to host family (red circle, Hipposideridae, green square, Vespertilionidae, blue triangle, Rhinolophidae).

Click here for additional data file.

Figure S3 UniFrac distances for pairwise comparisons among groups

X-axis, pairwise comparisons among groups. Y-axis, UniFrac distances. Box borders represent upper and lower interquartile ranges. Red lines, whiskers, and “+” represent the median values, 1.5 times the interquartile range beyond upper and lower quartiles, and outliers respectively. Significant differences in the UniFrac distances for pairwise comparisons among groups are shown in Table 2. WFVs, WFRf, WFHa, FVs, FRf and FHa are defined in the legend for Fig. 1. If the distance between two groups is significantly greater than that within the groups, the difference between these groups is significant.

Click here for additional data file.

Figure S4 Relative abundances of major taxa in wild bats’ fecal microbiotas at the phylum, family and genus levels

WFVs, WFRf and WFHa represent fecal samples from V. sinensis, R. ferrumequinum and H. armiger collected from the wild respectively.

Click here for additional data file.

Figure S4 Relative abundances of major taxa in captive bats’ fecal microbiotas at the phylum, family and genus levels

FVs, FRf and FHa represent fecal samples from captive V. sinensis, R. ferrumequinum and H. armiger respectively.

Click here for additional data file.

Data S1 Raw data per individual animal

Sex, age, weight, forearm length, and sample site information of each bat.

Click here for additional data file.

Data S2 Supplemental file 2

raw data of FHa1.

Click here for additional data file.

Data S3 Supplemental file 3

raw data of FHa1.

Click here for additional data file.

Data S4 Supplemental file 4

raw data of FHa2.

Click here for additional data file.

Data S5 Supplemental file 5

raw data of FHa2.

Click here for additional data file.

Data S6 Supplemental file 6

raw data of FHa3.

Click here for additional data file.

Data S7 Supplemental file 7

raw data of FHa3.

Click here for additional data file.

Data S8 Supplemental file 8

raw data of FHa4.

Click here for additional data file.

Data S9 Supplemental file 9

raw data of FHa4.

Click here for additional data file.

Data S10 Supplemental file 10

raw data of FHa5.

Click here for additional data file.

Data S11 Supplemental file 11

raw data of FHa5.

Click here for additional data file.

Data S12 Supplemental file 12

raw data of FHa6.

Click here for additional data file.

Data S13 Supplemental file 13

raw data of FHa6.

Click here for additional data file.

Data S14 Supplemental file 14

raw data of FHa7.

Click here for additional data file.

Data S15 Supplemental file15

raw data of FHa7.

Click here for additional data file.

Data S16 Supplemental file 16

raw data of FHa8.

Click here for additional data file.

Data S17 Supplemental file 17

raw data of FHa8.

Click here for additional data file.

Data 18 Supplemental file 18

raw data of FHa9.

Click here for additional data file.

Data S19 Supplemental file 19

raw data of FHa9.

Click here for additional data file.

Data S20 Supplemental file 20

Raw data of FHa10.

Click here for additional data file.

Data S21 Supplemental file 21

Raw data of FHa10.

Click here for additional data file.

Data S22 Supplemental file 22

Raw data of FRf1.

Click here for additional data file.

Data S23 Supplemental file 23

Raw data of FRf1.

Click here for additional data file.

Data S24 Supplemental file 24

Raw data of FRf2.

Click here for additional data file.

Data S25 Supplemental file 25

Raw data of FRf2.

Click here for additional data file.

Data S26 Supplemental file 26

Raw data of FRf3.

Click here for additional data file.

Data S27 Supplemental file 27

Raw data of FRf3.

Click here for additional data file.

Data S28 Supplemental file 28

Raw data of FRf4.

Click here for additional data file.

Data S29 Supplemental file 29

Raw data of FRf4.

Click here for additional data file.

Data S30 Supplemental file 30

Raw data of FRf5.

Click here for additional data file.

Data S31 Supplemental file 31

Raw data of FRf5.

Click here for additional data file.

Data S32 Supplemental file 32

Raw data of FRf6.

Click here for additional data file.

Data S33 Supplemental file 33

Raw data of FRf6.

Click here for additional data file.

Data S34 Supplemental file 34

Raw data of FRf7.

Click here for additional data file.

Data S35 Supplemental file 35

Raw data of FRf7.

Click here for additional data file.

Data S36 Supplemental file 36

Raw data of FRf8.

Click here for additional data file.

Data S37 Supplemental file 37

Raw data of FRf8.

Click here for additional data file.

Data S38 Supplemental file 38

Raw data of FRf9.

Click here for additional data file.

Data S39 Supplemental file 39

Raw data of FRf9.

Click here for additional data file.

Data S40 Supplemental file 40

Raw data of FRf10.

Click here for additional data file.

Data S41 Supplemental file 41

Raw data of FRf10.

Click here for additional data file.

Data S42 Supplemental file 42

Raw data of FVs1.

Click here for additional data file.

Data S43 Supplemental file 43

Raw data of FVs1.

Click here for additional data file.

Data S44 Supplemental file 44

Raw data of FVs2.

Click here for additional data file.

Data S45 Supplemental file 45

Raw data of FVs2.

Click here for additional data file.

Data S46 Supplemental file 46

Raw data of FVs3.

Click here for additional data file.

Data S47 Supplemental file 47

Raw data of FVs3.

Click here for additional data file.

Data S48 Supplemental file 48

Raw data of FVs4.

Click here for additional data file.

Data S49 Supplemental file 49

Raw data of FVs4.

Click here for additional data file.

Data S50 Supplemental file 50

Raw data of FVs5.

Click here for additional data file.

Data S51 Supplemental file 51

Raw data of FVs5.

Click here for additional data file.

Data S52 Supplemental file 52

Raw data of FVs6.

Click here for additional data file.

Data S53 Supplemental file 53

Raw data of FVs6.

Click here for additional data file.

Data S54 Supplemental file 54

Raw data of FVs7.

Click here for additional data file.

Data S55 Supplemental file 55

Raw data of FVs7.

Click here for additional data file.

Data S56 Supplemental file 56

Raw data of FVs8.

Click here for additional data file.

Data S57 Supplemental file 57

Raw data of FVs8.

Click here for additional data file.

Data S58 Supplemental file 58

Raw data of FVs9.

Click here for additional data file.

Data S59 Supplemental file 59

Raw data of FVs9.

Click here for additional data file.

Data S60 Supplemental file 60

Raw data of FVs10.

Click here for additional data file.

Data S61 Supplemental file 61

Raw data of FVs10.

Click here for additional data file.

Data S62 Supplemental file 62

Raw data of WFHa1.

Click here for additional data file.

Data S63 Supplemental file 63

Raw data of WFHa1.

Click here for additional data file.

Data S64 Supplemental file 64

Raw data of WFHa2.

Click here for additional data file.

Data S65 Supplemental file 65

Raw data of WFHa2.

Click here for additional data file.

Data S66 Supplemental file 66

Raw data of WFHa3.

Click here for additional data file.

Data S67 Supplemental file 67

Raw data of WFHa3.

Click here for additional data file.

Data S68 Supplemental file 68

Raw data of WFHa4.

Click here for additional data file.

Data S69 Supplemental file 69

Raw data of WFHa4.

Click here for additional data file.

Data S70 Supplemental file 70

Raw data of WFHa7.

Click here for additional data file.

Data S71 Supplemental file 71

Raw data of WFHa7.

Click here for additional data file.

Data S72 Supplemental file 72

Raw data of WFHa8.

Click here for additional data file.

Data S73 Supplemental file 73

Raw data of WFHa8.

Click here for additional data file.

Data S74 Supplemental file 74

Raw data of WFHa9.

Click here for additional data file.

Data S75 Supplemental file 75

Raw data of WFHa9.

Click here for additional data file.

Data S76 Supplemental file 76

Raw data of WFHa10.

Click here for additional data file.

Data S77 Supplemental file 77

Raw data of WFHa10.

Click here for additional data file.

Data S78 Supplemental file 78

Raw data of WFRf1.

Click here for additional data file.

Data S79 Supplemental file 79

Raw data of WFRf1.

Click here for additional data file.

Data S80 Supplemental file 80

Raw data of WFRf14.

Click here for additional data file.

Data S81 Supplemental file 81

Raw data of WFRf14.

Click here for additional data file.

Data S82 Supplemental file 82

Raw data of WFRf24.

Click here for additional data file.

Data S83 Supplemental file 83

Raw data of WFRf24.

Click here for additional data file.

Data S84 Supplemental file 84

Raw data of WFRf25.

Click here for additional data file.

Data S85 Supplemental file 85

Raw data of WFRf25.

Click here for additional data file.

Data S86 Supplemental file 86

Raw data of WFRf26.

Click here for additional data file.

Data S87 Supplemental file 87

Raw data of WFRf26.

Click here for additional data file.

Data S88 Supplemental file 88

Raw data of WFRf27.

Click here for additional data file.

Data S89 Supplemental file 89

Raw data of WFRf27.

Click here for additional data file.

Data S90 Supplemental file 90

Raw data of WFRf28.

Click here for additional data file.

Data S91 Supplemental file 91

Raw data of WFRf28.

Click here for additional data file.

Data S92 Supplemental file 92

Raw data of WFRf29.

Click here for additional data file.

Data S93 Supplemental file 93

Raw data of WFRf29.

Click here for additional data file.

Data S94 Supplemental file 94

Raw data of WFVs1.

Click here for additional data file.

Data S95 Supplemental file 95

Raw data of WFVs1.

Click here for additional data file.

Data S96 Supplemental file 96

Raw data of WFVs3.

Click here for additional data file.

Data S97 Supplemental file 97

Raw data of WFVs3.

Click here for additional data file.

Data S98 Supplemental file 98

Raw data of WFVs4.

Click here for additional data file.

Data S99 Supplemental file 99

Raw data of WFVs4.

Click here for additional data file.

Data S100 Supplemental file 100

Raw data of WFVs6.

Click here for additional data file.

Data S101 Supplemental file 101

Raw data of WFVs6.

Click here for additional data file.

Data S102 Supplemental file 102

Raw data of WFVs7.

Click here for additional data file.

Data S103 Supplemental file 103

Raw data of WFVs7.

Click here for additional data file.

Data S104 Supplemental file 104

Raw data of WFVs8.

Click here for additional data file.

Data S105 Supplemental file 105

Raw data of WFVs8.

Click here for additional data file.

Data S106 Supplemental file 106

Raw data of WFVs10.

Click here for additional data file.

Data 107 Supplemental file 107

Raw data of WFVs10.

Click here for additional data file.

We are grateful to Lixin Gong, Biye Shi, Zhongwei Yin and Chuantao Song for their great contributions to samples collection. Sequencing service was provided by Personal Biotechnology Co., Ltd. Shanghai, China. We thank Traci Raley, MS, ELS, from Liwen Bianji, Edanz Editing China for editing a draft of this manuscript.

Additional Information and Declarations

Competing Interests

Author Contributions

Animal Ethics

Field Study Permissions

DNA Deposition

Data Availability

The authors declare there are no competing interests.

Yanhong Xiao conceived and designed the experiments, performed the experiments, analyzed the data, prepared figures and/or tables, authored or reviewed drafts of the paper, approved the final draft.

Guohong Xiao performed the experiments, prepared figures and/or tables, approved the final draft.

Heng Liu, Xin Zhao, Congnan Sun and Xiao Tan performed the experiments, approved the final draft.

Keping Sun authored or reviewed drafts of the paper, approved the final draft.

Sen Liu performed the experiments, approved the final draft.

Jiang Feng conceived and designed the experiments, contributed reagents/materials/analysis tools, authored or reviewed drafts of the paper, approved the final draft.

The following information was supplied relating to ethical approvals (i.e., approving body and any reference numbers):

All research carried out in this study was approved by the National Animal Research Authority of Northeast Normal University, China (NENU-20080416).

The following information was supplied relating to field study approvals (i.e., approving body and any reference numbers):

Sampling was conducted with the permission of the Forestry Bureau of Jilin Province of China.

The following information was supplied regarding the deposition of DNA sequences:

All raw sequences were deposited in the NCBI Sequence Read Archive under accession number SRR8238420–SRR8238472.

The following information was supplied regarding data availability:

All raw sequences are available in Files S9–S109.

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
