# Peer review of "Captivity causes taxonomic and functional convergence of gut microbial communities in bats"

_PeerJ, doi:10.7717/peerj.6844_

## Round 0.1 · original submission · Major Revisions

Your paper has now been assessed by two expert reviewers. From my own reading of the paper, I agree with both reviewers that this paper has some interesting findings and addresses an important question. However there are a number of issues which need to be addressed before the paper can be recommended for publication. Both reviewers have provided very detailed reviews which should help you revise your manuscript. In addition, I raise the following suggestions:

Comments:

Line 161-166: Although your sequencing depth was higher for captive bats, I’m still surprised that you detected roughly double the number of OTUs for what should have been a greatly simplified gut microbiome due to the single-food-source diet and captive conditions. You wait until the last line of discussion to mention this, but I think it requires much stronger treatment. Reviewer 1 has provided some suggestions in this regard which you should address. I would also suggest you include some additional references from other taxa that demonstrate this phenomenon, rather than just focussing on the rodent example e.g. Kueneman et al 2016 Proc B 20161553; and perhaps: van Schooten B, Godoy-Vitorino F, McMillan WO, Papa R. 2018. Conserved microbiota among young Heliconius butterfly species. PeerJ 6:e5502 https://doi.org/10.7717/peerj.5502

Line 165: You should also define what you mean by ‘all enough’

Line 180 / Line 216-220: I have a bit of an issue with the framing of the paper with repsect to repeated assertion that you fed the bats identical diets. In fact, only 2 of 3 bat species were fed the same diet, but the paper reads as if all 3 received the same diet. This is something that has been noted by the reviewers, and so I think your paper needs to be reframed slightly to acknowledge the actual design of your study. Unfortunately, because of your approach diet and species are confounded for the larger bat species, so you need to be careful about your interpretations.

Line 188: A PCoA doesn’t work at the level of phylum, but at the OTU level (rows of your abundance matrix). The PCoA in fact has no idea what taxonomy any of the OTUs are.

Line 245-246: Perhaps the nutrient composition of the two larvae are similar, but that depends on what level you assess nutrition i.e. macronutrients such as fat and protein vs micronutrients like antioxidants. In addition, similar macronutrient profiles does not guarantee similar effects on gut microbiome. You have some evidence for this, but unfortunately cannot say with certainty because you didn’t cross species with diet.

Fig. 1: The Y axis is labelled “Observed Species” but I think you mean ‘OTUs’ as per the figure legend. You also don’t define what your error bars are

Fig 2, 4, 5: You don’t define the abbreviations of the venn diagram sections / bat species in Figs. 2 and 5. Nor the X axis labels in Fig. 4

Reviewer 1 ·

Basic reporting

1. Basic Reporting

a. Clear/unambiguous English:

The English should be improved for clarity and comprehension throughout the manuscript; example areas that require improvement are lines 22-23, 48-49, 273-274. Current phrasing is awkward and makes comprehension of certain points difficult. The title also could use improvement; the authors did not raise the bats from pups, so the title should be amended to something like “Captivity Causes Taxonomic and Functional Convergence of Gut Microbial Communities in Bats”.

b. Literature references, sufficient context:

Line 42: Consider also that the social behavior of animals that live in close proximity, such as bats, can drive gut microbiota structure in the wild (Tung et al. 2015; Moeller et al. 2016; Kolodny et al. 2018). This has especially been demonstrated in bats roosting together. As such, you should add a sentence explaining whether or not you housed bats in the same enclosure during the controlled feeding portion of the study.

Line 41-42: Rather than saying “host-specific”, you should expand this part of the introduction to discuss that in addition to diet, host evolutionary history has a strong influence on gut microbiome composition (Ley et al. 2008; Phillips et al. 2012; Ingala et al. 2018).

Line 45-47: Specify that the taxonomic composition of gut microbial communities converge in the references you cited.

Line 57-58: You should qualify what you mean by “herbivores are only a portion of mammals” by specifying what percentage of mammals are herbivorous. You might also state that insectivory is thought to be the ancestral feeding condition for placental mammals (O’Leary et al. 2013), as well as for bats (Dawson and Krishtalka 1984), which makes understanding the microbiome of insectivorous animals important from an evolutionary perspective.

Line 59-60: What is interesting about this study compared to (Kohl et al. 2014) is that you are studying three bat species that are members of different families (Hipposideridae, Rhinolophidae, and Vespertilionidae). Kohl et al. 2014 studied two closely related woodrats (Neotoma spp.). You might better frame your research question more like this: “Given the phylogenetic distance among hosts studied here, it is unclear whether homogenization of diet/environment or host evolutionary history has a stronger impact on microbiome community composition.”

Line 77-78: In the materials and methods, the authors describe performing functional prediction on the bat microbial communities, but do not contextualize why this is important or relevant in the introduction, nor do they hypothesize about what they expect to find. I suggest further developing the introduction by discussing that taxonomy and function are decoupled in microbial ecosystems (Graham et al. 2016; Louca et al 2016; Inkpen et al. 2017). There is also a study in insectivorous bats which showed that among three different bat species, microbial community taxonomic composition varied but all were functionally convergent (Phillips et al. 2017). This context would help the authors articulate a hypothesis about the functional assays.

Line 81-94: There are several ways to sample gut microbiota (swabs, guano, intestinal sections). You should justify why you chose to sample feces only. A helpful reference is (Ingala et al. 2018), which shows that signal of diet in the microbiome is more detectable in fecal samples than intestinal samples.

Line 90: You specify that you collected fecal samples from individuals “after more than two months”, which makes it sound as if you collected two month old fecal pellets. Rephrase for clarity and state that you kept the bats on a homogenous diet for two months, and collected fecal pellets from captive animals X hours/days after defecation.

Line 110-111: Please cite the appropriate reference for primers 338F and 806R.

Line 132-134: Did you use open- or closed-reference clustering method? Please specify

Line 145: Please state the version of QIIME you used, and please cite individual R packages when you mention them later in this section, including the version (i.e. phyloseq v 1.1-2, McMurdie & Holmes 2013, etc.). Version of PICRUSt is also missing.

Lines 206 & 208: KEGG should be capitalized (Kyoto Encyclopedia of Genes & Genomes)

Line 273-281: It is interesting that you found microbial diversity increased after bats were taken into captivity. It would be good to provide some hypothesis or speculation as to why this might be; for example, is it possible that the bat species were exposed to one another’s microbes (thus explaining taxonomic homogenization), and so the overall community becomes more diverse from sharing among species? Alternatively, could contaminating laboratory bacteria be the cause?

c. Professional article structure, raw data shared:
Article structure appears standard and professional. Raw data is shared as supplementary files, and clicking through a few, the downloads work successfully. Raw data was also uploaded to NCBI. You may also consider making your R code available for easier replication.

d. Self-contained with relevant results to hypothesis:
Paper is self-contained and results follow hypotheses logically.

Experimental design

2. Experimental Design

a. Original research within aims and scope?:
The manuscript is within the aims and scope of PeerJ and meets criteria for original research

b. Research question well defined, relevant, meaningful?:
The research question is somewhat well-defined, but it would be more compelling if the authors addressed several contextual comments made above. In particular, they should better introduce the functional aspect of the study, which isn’t introduced to the reader until the methods section.

c. Rigorous investigation performed to high technical/ethical standard?:
All ethical standards seem adequate, and appropriate permissions were obtained for the fieldwork. Investigation was performed using techniques that are standard and accepted in the microbiome field.

Line 149-151: Please check the assumptions of the Student’s t- test; this test assumes a normal distribution of data and homoscedasticity of variance. Please check that your Unifrac distances do not violate these assumptions, in which case the non-parametric Wilcoxon test would be more appropriate (Xia and Sun 2017). In addition, in Table 2 it appears you use the Bonferroni post-hoc correction method, but you do not mention this in the Methods. Please add to methods for clarity.

Your results regarding the functional convergence are very interesting, but I am curious why you only tested presence/absence of functions and not relative abundance of functions. For example, all captive bats may converge on the same functions in terms of presence/absence, but the relative abundance of functions may differ substantially among host species. See such an example in Phillips et al. 2017 (Integrative & Comparative Biology): “Given that comparisons were made among insectivorous host species, frequency differences for metagenome functions pertaining to macromolecule metabolism may reflect host lineage-specific metabolic fine-tuning.” Please explain if any of the hosts differed in terms of relative abundance of functions.

Figure 3: In the figure legend, you indicate the shape of symbols correspond to the “host order”, but all bats belong to the Order Chiroptera. I think what you meant here is “host family”. In any case, there is no clear indication of what the families are because the legend shapes are just the Species names. Please correct the legend to show the appropriate host families (i.e. Red circle = Hipposideridae, green square = Vespertilionidae, blue triangle = Rhinolophidae)

d. Methods described with sufficient detail & information to replicate?:
Thank you for describing the methods in good detail. Metadata and raw sequences are provided so that analysis can be replicated.

Validity of the findings

3. Validity of Findings

a. Impact and novelty not assessed. Negative/inconclusive results accepted. Meaningful replication encouraged?
The authors encourage replication of their study. Results seem overall compelling and deserve future replication.

b. Data is robust, statistically sound, & controlled
The data was robust, and treated in an overall statistically sound manner. As far as controls, the authors do not mention whether a negative extraction or PCR control was sequenced alongside their fecal samples. Many kits have bacterial contaminants (Salter et al 2014) that can be filtered from microbiome samples in silico. The R package “decontam” can be used to filter out potential contaminants using either (a) a negative control sample, or (b) the initial concentrations of each library.

One analysis missing from the manuscript is a bar plot of overall microbiome composition. It would be helpful as an additional control to have a supplemental figure of sample by sample microbiome composition, and a brief discussion if the community compositions are consistent with previous work on bat microbiomes (Phillips et al. 2012, Carrillo-Araujo et al. 2015, Phillips et al. 2017, Ingala et al. 2018).

c. Conclusions are well-stated, linked to research question and limited to supporting results
Conclusions drawn from the analyses are easy to interpret and directly relate to the research questions and results.

d. Speculation is welcome but should be clearly identified.
Speculation in the discussion is limited and appropriate.

Additional comments

This manuscript seeks to answer whether homogenization of diet and environment leads to convergence of gut microbial communities in bats. Overall, the manuscript is interesting, technically sound, and uses appropriate methods to answer the question. After substantial revision of the English language, incorporation of some needed context and references, and addition of a few critical elements in the results, this manuscript will be in good shape.

Reviewer 2 ·

Basic reporting

Whilst I appreciate that English is likely not the first language of the authors, the standard of English throughout the manuscript requires significant attention. The authors should consider having the manuscript thoroughly revised by a colleague who is a native English speaker or consulting a professional editing service.

In general, I find that the manuscript is well supported by reference to prior literature. The figures are also of a good standard, although the legends should be made more informative if the manuscript is revised. However, there are some structural issues that I believe need to be addressed.

Lines 49 – 57 of the introduction discusses the findings of Kohl et al (2014) at great length. Whilst obviously relevant, this work could be summarised much more succinctly to improve the flow of the manuscript.

The methods section which relates to sample collection (lines 81-94) lacks detail on potentially important aspects of the study. Most of the information can be gathered from table 1, however it should be at least mentioned in the main body of the text for clarity. For example, from how many sites were the samples for each bat species collected? (Since there are likely site-specific gut microbiomes within each species group). At what time of year were the samples collected (Since bat diet and feeding ecology vary seasonally).

Experimental design

The hypotheses that this work seeks to test are well defined, however the experimental design is weak.

My largest concern is that given the methodologies as they are currently presented, what the authors have categorised is the faecal microbiome as a whole, rather than the host-associated gut microbiota of the sampled bat species. Aside from an abundance filter, there is little attempt to ascertain what detected bacterial species are truly host-derived and which are transient environmental bacteria or commensals of the prey animals. Equally, despite mentioning inter-species variation in diet and intra-species seasonal variation in diet composition in the introduction, the authors seem to have given little thought to the actual composition of the diet of the wild bats that were sampled. Given that the structure of the host gut microbiota cannot be teased apart from the faecal microbiome, the authors cannot exclude the possibility that observed divergence in faecal microbiome structure between wild bats and observed convergence between the faecal microbiome structure of captive animals are not entirely due to the absence of different environmental and prey-derived transient microbes in the captive setting. It is probably not possible to address this issue completely without undertaking a much larger study incorporating dietary classification via metabarcoding and classification of the microbiome of the invertebrate prey.

The project would undoubtedly benefit from some further sequencing to help address these concerns. However, if this is categorically not an option then the authors should add extensive caveats to the discussion that interpret the findings of the paper in light of this uncertainty.

My secondary concerns which should be addressed subject to correction of the first are;
• While common place in the literature, the binning of OTUs based on universally applied cut off of 97% identity is likely inappropriate for some of the detected species and somewhat antiquated in light of newer clustering algorithms. The authors should consider using DADA2 to define bacterial “species”.
• Why did the authors decide to use UniFrac distances over other distance metrics? A brief discussion of this in the methods would be helpful.

Validity of the findings

Given my primary concern under the experimental design section, I believe that the findings of this paper as they are currently presented are invalid.

To reiterate, I believe that what the authors have measured is the composition of the faecal microbiome of wild vs captive bats. Whilst they have found that the composition of these microbial communities converges between species fed on a uniform diet, the authors cannot conclusively say that this convergence is due to changes in the composition of the host-associated gut microbiome rather than a much larger shared component of the faecal microbiome based on a completely shared diet comprising of meal worms and their commensal bacteria.

If the authors cannot address this uncertainty through additional sequencing, then they should incorporate extensive caveats and discuss their results in light of this uncertainty.

Finally, there is no mention that the data generated by this study has been or will be made publicly available.

Additional comments

In this paper Yanhong Xiao et al. seek to use a metabarcoding approach to categorise and compare the gut microbial communities of three bat species both in captivity and in the wild. Unfortunately, I have several concerns regarding the methodologies of the work and the quality of the resulting manuscript that must be addressed before the manuscript can be accepted for publication. However, should the authors comprehensively review and improve the manuscript then I believe the study could act as a starting point or useful reference for others working in similar areas.

This review focusses on the over-arching problems which I believe require attention rather than specific issues line by line.

---

## Round 0.2 · Minor Revisions

Your manuscript has now been reassessed by one of the original expert reviewers, whose comments are appended below.

Both the reviewer and I agree that the manuscript has improved greatly with revisions. The reviewer has also attached a markup version of the manuscript with suggestions for stylistic changes which I encourage you to consider carefully.

Regarding the reviewer's comments about figures, I agree that some adjustments are needed. For example, I request that the rarefaction curves should be moved to supplementary material (and abbreviations redefined in the legend of figure 2). Also the 3D panels A and B in figure 3 are difficult to interpret (e.g. it looks like some data are outside the axis limits of PC2 because of the perspective), and actually duplicate information in panels C and D, which are much clearer and easy to interpret. Likewise, Fig 4 is not that useful with respect to your manuscript and should be moved to supplementary material.

I hope these comments will be helpful for the revision of your manuscript.

Reviewer 2 ·

Basic reporting

My comments regarding the standard of the English and the flow of the introduction have been satisfactorily addressed.

However, some small errors remain and I have highlighted all of these that I noticed in the attached version of the tracked changes document.

The methods section is improved for the additional details provided

A few additional comments

On lines 67 - 68 the authors suggest that captive animals are usually housed in a uniform manner. Could this be clarified, and if it is the intention of the authors to suggest that different species are kept under uniform conditions at zoos etc could this be backed up with evidence? I would suggest that most responsible zoos endeavor to mirror the conditions of the native habitat of their captive species as closely as possible in many instances.

The section of the introduction between lines 96 and 107 is very methods heavy and any information contained within this portion which in not already in the methods section should be moved there instead.

The first section of the methods (lines 127 to 159) is very difficult to follow and should be split into separate "Field" and "Methods" sections.

The final section of the results (lines 287 to 310) is also very difficult to read and should be simplified or clarified to improve comprehension.

I also believe that the manuscript potentially contains far to many figures for a manuscript of this type and length. The authors should consider keeping 2 or 3 main figures and relegating the rest to supplementary material.

Experimental design

I am disappointed that the authors did not deem it possible to address some of my major concerns regarding experimental validity. However, I find the inclusion of extensive caveats an acceptable solution in this instance.

I would also suggest to the authors that just because something is the most widely used example of a given tool, does not mean that those who are aware of problems surround that tool should continue to use it and so prevent the collective forward progression of the field. In future studies, a move away from arbitrary 97% cut off OTU clustering would be beneficial to all.

Validity of the findings

No comment

Additional comments

The extensive revisions that you made to the manuscript have greatly improved it for the last version. However I believe there are still some minor issues that need to be addressed prior to publication. Please see my above comments and attached version of the track changes document.

Annotated reviews are not available for download in order to protect the identity of reviewers who chose to remain anonymous.

---

## Round 0.3 · accepted · Accept

Thank you for making the requested changes to the manuscript. Having assessed the revisions, I am now happy to accept the paper.

Please note for the proof-reading step, on line 53 of the track changes version of the manuscript I believe the term should be 'specific microbes'.

#